# Cost-Effective Droplet Generator for Portable Bio-Applications

**DOI:** 10.3390/mi14020466

**Published:** 2023-02-17

**Authors:** Lin Du, Yuxin Li, Jie Wang, Zijian Zhou, Tian Lan, Dalei Jing, Wenming Wu, Jia Zhou

**Affiliations:** 1School of Mechanical Engineering, University of Shanghai for Science and Technology, Shanghai 200433, China; 2School of Optical-Electrical and Computer Engineering, University of Shanghai for Science and Technology, Shanghai 200433, China; 3Institute of Biological and Medical Engineering, Guangdong Academy of Sciences, Guangzhou 510075, China; 4State Key Laboratory of ASIC and System, School of Microelectronics, Fudan University, Shanghai 200433, China

**Keywords:** lab chip, microdroplet, squeezing model, capillary number

## Abstract

The convenient division of aqueous samples into droplets is necessary for many biochemical and medical analysis applications. In this article, we propose the design of a cost-effective droplet generator for potential bio-chemical application, featuring two symmetric tubes. The new droplet generator revisits the relationship between capillary components and liquid flow rates. The size of generated droplets by prototype depends only on generator dimensions, without precisely needing to control external flow conditions or driving pressure, even when the relative extreme difference in flow rate for generating nL level droplets is over 57.79%, and the relative standard deviation (RSD) of the volume of droplets is barely about 9.80%. A dropper working as a pressure resource is used to verify the rapidity and robustness of this principle of droplet generation, which shows great potential for a wide range of droplet-based applications.

## 1. Introduction

Droplet microfluidics are a technology that have been developed for precise control and manipulation of microscale fluids in past decades [1,2]. Microdroplets are widely used in biochemical analysis such as polymerase chain reaction (PCR) [3,4], multiple emulsions drug delivery [5,6], single-cell analysis [7,8], and microreactors [9]. The principle of the droplet generator can be roughly divided into two directions: continuous flow and continuous wetting. Producing droplets described in a continuous tube is a common droplet preparation method as it is affordable and has controllable components. Surface tension and viscous shear primarily affect the ratio between capillary and shear forces, which was systematically studied [10,11,12,13]. A common method for resizing capillary numbers is to resize capillary components, such as the inner diameter of the tube. These expandable sections can improve diversification of droplet size and multiple emulsion forms, which is promising in various applications just as antimicrobial susceptibility testing [14]. However, in the squeezing model theory, droplet sizes are determined only by the ratio of the flow rates of two immiscible liquids in microcontinents. This droplet generating in relation between droplet size and flow rates makes the squeezing regime attractive for applications. Efficient use of this technique requires a strictly limited specifications of components and a good understanding of limits of the squeezing regime. It is known that these droplets dynamic behavior strongly depends on their size, while the dynamics in forming droplets offer rich phenomenological complexity [15]. In portable bio-applications, repeatable and rapid experiments require precise droplet generation techniques to ensure the reliability of analysis results. A variety of droplet on-demand generators [16,17,18,19,20] are described in former literature, from the control accuracy of fluid pressure to precision-manufactured tube structure. Continuous microchannel and sealed chamber chips produce highly monodispersed droplet streams and microdroplets by using polymeric microtubes [16], flow-focusing geometry [17], multilayer geometry [18], spinning conical frustum [19], and chamber formats [20]. The performance of droplet generated may vary with the precise conditions of the equipment and the parts included in the generators, such as precision-matched chips with necessary extra bonding or sealing. However, most existing designs of droplet generators are either complex, costly, or difficult to adapt to control the ratio between shear and capillary forces, especially in portable bio-application. Usually, high precision and reproducibility of droplet generators are required in combination with independence on material parameters of two liquids [10,21]. Additionally, a traditional flow-focusing structure droplet generator requires more than two micropumps to control the flow and timing precisely to ensure the ratio of flow rates of two immiscible liquids, independently of capillary number, as described by squeezing model. The volume of these droplets can be adjusted by changing the rates of pump flow of dispersed phase and continuous phase [15,22]. The pinch-off process that generates droplets is regulated by the inflow of liquid into an orifice. The squeezing model was commonly accepted for applications, but does not account for cost-effective assurance of the ratio between continuous liquids while the droplet is formed. The sample reagents in the pressure control tube are often consumptive, deteriorating detection accuracy. Many systems are designed to produce droplets with volumes in nL, which frequently challenges the range of volume control accuracy in point-of-care testing [23,24]. As the characteristic scale decreases, the surface tension of the droplet gradually exceeds mass force and turns to the most important microscale [25,26], which dampens the production of droplets. The situation will be more unpredictable if the properties of the dispersed phase are unknown [27,28].

This article proposes a cost-effective droplet generator with only one set of inlet–outlet, which is highly reproducible and leads to the generation of monodisperse droplets in a flow-focusing device. By constructing symmetry tubes to reduce dependence on absolute pressure value and manufacturing accuracy, this device offers a simple, reliable, and high-precision property for generating droplets. Additionally, with a wide flow rate range for generating droplets, the process of generating droplets is robust. So, the pump can be replaced with a simpler pressure device, for example, a dropper. The device can also produce uniform droplets with even a finger as a power source. Two symmetric tubes were used to make sure the pressure at the ends of the configuration phases was equal to the squeezing model theory. A wide range of pump flow rates can be used in droplet preparation process, which benefits from the shared pressure of the symmetric tube. In this way, we carried out the experiments.

## 2. Droplet Generator Principles

The droplet generator is composed of two symmetric tubes, shared main tubes, one pump, and two flow junctions. The device comprises two T-junctions of inlet–outlet channels that are connected to symmetric tubes and deliver two immiscible liquids. Surface tension and viscous shear are influenced by direct-controlled “capillary-dominated” components in the microcontinent. Meanwhile, the Poiseuille Law [29,30,31] dictates that the pressure applied to symmetric tubes maintains the flow rate stable. Squeezing does not affect the ratio of flows between immiscible liquids sensitively. Two parts of liquid are stored in tubes, and terminal pressures of these two liquids will always keep the same value in the inlet and outlet of the tubes, respectively, which promises relative stability for producing microdroplets in junction compared with conventional ones. The pressure difference arises from the sum of viscous flow inside the dispersed phase and the difference in curvature of the interface at different size capillary tubes. Once the outlet is provided with a threshold negative pressure by a pump, the liquid will gradually start flowing, under the pressure gradient. The dispersed phase is pulled into the junction forming droplets under shear force from a continuous phase. The pump is the collector of produced microdroplets, which can conveniently be moved anywhere for further applications.

The theoretical framework in forming droplets starts with relative magnitudes of surface tension and viscous shear, as captured by the capillary number, and is in contrast to the viscosities between the two phases. The schematic of the droplet generator developed in the present work is given in Figure 1, distinct “visco-capillary” regimes have been identified in this type of flow-focusing structure (Figure 1a(i)–(iii)) by other scientists. In this model, two symmetric circular pipes are constructed and interconnected with each other, with a small-sized capillary tube connected to generate droplets at the end-intersection of two pipes. Filled on one side is a continuous liquid phase, and on the other side is a dispersed phase. During the transportation of continuous phase and dispersed phase, pressure drops at the two ends of the system should remain the same. Based on Poiseuille Law, the relationship between rate of flow and the difference in pressure at both ends of the tube can be expressed by $Q=\frac{1}{8\eta L}R^{4}(P_{2}-P_{1})$. The radius and length of tube are R and L, and the viscosity of fluid is η. It can be seen from the formula that Q is proportional to pump pressure and four power of R. The total flow is the sum of flows in the two tubes, which can be expressed by $\left(Q=Q_{1}-Q_{2}\right)$. Then, the proportion of droplets in the total liquid can be derived into the following: $\frac{Q_{2}}{Q}=\frac{\eta_{1}L_{1}}{\frac{R_{1}}{R_{2}}\eta_{2}L_{2}+\eta_{1}L_{1}}$. It can be seen from the above formula that the ratio of dispersed/continuous phase can be changed by controlling the pipe diameter with only a mono-pump, and it is immune from the influence of pump pressure. In the schematic of the droplet generator shown in Figure 1b, characters (i), (ii), and (iii) in the circle indicate effective differences in pipe diameter. Once the direct-controlled “capillary-dominated” components, such as relative magnitude of surface tension and shear, and the viscosities between the two phases, are established, the simple relationship between droplet size and flow rate are definite as well. Figure 1a shows the dynamic process of the droplet generator in the preparation of droplets, (i) indicates the dimensional relationship in the principles of Poiseuille law, (ii) indicates that the water phase liquid has just reached the fluid tangent position, (iii) indicates that the liquid is subjected to the oil phase under the action of fluid shear force forming a droplet.

As a consequence of capillary tubes, continuous phase size and dispersed phase size are different, as are pressure differences that cause flow. Therefore, it is essential to produce droplets only when pressure difference is sufficient to overcome pressure loss of both parts of the fluid. Adopting the electric–hydraulic analogy, we describe the accumulation of the droplet akin to electric RC circuits: only if the pump flow rate exceeds critical value, the fluids in two tubes can flow at the same time to generate droplets. So, the droplets produced by different pipe diameters have different minimum flow pressure differences. Only when critical flow rate is exceeded, the two-phase flow rate can generate a stable droplet whose size is consistent with flow rate.

## 3. Experiment Setups

The tubes were designed by applying four silicone tubes (inner diameter 3 mm, outer diameter 4 mm). The junctions were made up of two T-junctions (inner diameter 0.8 mm), and one of the T-junctions contains a Teflon capillary tube shown in Figure 2a,b. A Teflon capillary tube with different diameters was made of PTFE (inner diameter 0.3 mm, outer diameter 0.6 mm), which was produced by drawing a hot stretching process under a heat gun and cutting it at different lengths into different inner diameters at the ends. More detail will be found in Appendix B, Figure A1. The value of the flow rate was controlled by a 10 mL syringe assembled in a pump (KDS-210, KD Scientific, Holliston, MA, USA). Glue was used to seal the joints. The materials mentioned above were connected according to the structure shown in Figure 2c. The water phase was simulated by aqueous red ink (Ruwen, Shanghai, China), and the oil phase was simulated by mineral oil (1864006, Bio-Rad Laboratories, Inc., Hercules, CA, USA). The characteristics of the components of droplet generator are shown in Table 1.

Firstly, the Teflon capillary tube was embedded inside a T-junction, with some glue to seal the outside of thicker tube with the inside of the T-junction. Then, aqueous red ink and mineral oil were injected into two silicone tubes, respectively. To generate droplets smoothly, we connected the ink-filled silicone tube to the end of the T-junction near the thicker Teflon tube. Another silicone tube filled with mineral oil was connected to a T-junction as a shared tube. The transfer-connected operation should carefully avoid liquid overflow and backflow caused by an excessive external force. The last silicone tube was combined with a T-junction and syringe. The pump worked at the end of the last silicone tube to offer negative pressure. The situation of droplet generation inside the T-junction was observed by a camera, recording the morphology and flow rate of the droplets. Drop sizes are analyzed using image processing software ImageJ.

## 4. Results and Discussions

The dynamics of droplet formation are generally governed by surface tension and viscous shear, and a range of control parameters that influence the generation of a single droplet has been studied [10]. This article focuses on the mechanisms of portable droplet generation in symmetric tubes. From theoretical analysis, droplet volume depends on the Teflon capillary tube size. Using a droplet generator with different capillary tube sizes shown in Figure 2b, four droplets of different sizes were produced. The droplet volumes were about 400 nL, 60 nL, 30 nL, and 5 nL, respectively, and the droplet pictures are shown in Figure 3a. This is consistent with the prediction of theoretical formula in Section 2, stating that the size of droplets can be precisely controlled by selecting the appropriate diameter of Teflon capillary tube.

From the theoretical model in Section 2, in order to generate droplets stably, the system should overcome different pressure differences from continuous phase and dispersed phase in the tube. Thus, there is a critical flow rate existing for stable generation of droplets. Figure 3b summarizes the critical flow rate of a droplet generator for four kinds of droplets. As shown in Figure 3b, the critical flow rate is 0.15 mL/min, 0.13 mL/min, 0.07 mL/min, and 0.05 mL/min, respectively, for four kinds of droplets with the volume of 5 nL, 30 nL, 60 nL, and 400 nL, which means a larger critical flow rate is necessary to generate smaller droplets. The broad range of flow rate in generating 30 nL droplets is up to more than 0.87 mL/min (see Appendix B, Figure A2). The relationship of production rate of droplets against the pump flow rate shows that the frequency of droplet generation increases with the increase in pump speed. This phenomenon can be analyzed by applying the size of capillary tube. According to the theoretical model in Section 2, the thinner the capillary tube is, the smaller the results $Q_{2}/Q$ will achieve, and a larger pressure drop will be produced. Meanwhile, the thinner capillary tube indicates a larger difference between pressure drops in continuous phase and dispersed phase. Thus, we can find droplet size is inversely proportional to critical flow rate. When the pump flow rate is too high, the speed of droplet generation will be too fast, which will lead to blurred droplets or integrated droplets (see Figure A2 from Appendix B). The droplets with a volume of 30 nL cannot be distinguished from each other in the image when the velocity is above 0.29 mL/min. The droplets with a volume of 60 nL were integrated into a larger droplet when the velocity was above 0.16 mL/min. The droplet with a volume of 30 nL has an RSD of 7.86% at a flow rate of 0.14 mL/min, an RSD of 5.64% at a flow rate of 0.23 mL/min, and an RSD of 7.72% at a flow rate of 0.31 mL/min. The droplet with a volume of 60 nL has an RSD of 6.36% at a flow rate of 0.08 mL/min, an RSD of 7.51% at a flow rate of 0.11 mL/min, and an RSD of 7.07% at a flow rate of 0.15 mL/min. The droplet with a volume of 400 nL has an RSD of 5.06% at a flow rate of 0.07 mL/min, an RSD of 3.98% at a flow rate of 0.12 mL/min, and an RSD of 7.91% at a flow rate of 0.17 mL/min.

We further investigated the effect of a mono-pump flow rate on droplet size when the flow rate reaches the critical value to generate the droplets stably. Taking the droplet with a volume of 60 nL as an example, Figure 4a gives an evolution of droplet size with an increasing flow rate of a mono pump from 0 mL/min. As discussed above, the droplet with a volume of 60 nL will be produced when the flow rate of the mono pump reaches its critical value of 0.07 mL/min. As the flow rate goes up to 0.16 mL/min, the droplet size generally remains constant. And the corresponding relative standard deviation (RSD) is 9.02%. The finding shows the volume of droplets produced from the portable approach is homogenous. Additionally, the droplets can be produced at different speeds, reducing dependence on high-precision external pumping equipment. Further, Figure 4b summarizes the RSD of the volume of four kinds of droplets produced in Figure 3, and corresponding RSD for these four kinds of droplets is 8.76%, 11.94%, 9.02%, and 6.70%, respectively, when changing the flow rate of the mono-pump. This phenomenon can still be analyzed using the theoretical model in Section 2. It can be found that for a droplet generator with fixed structural sizes, the flow rate ratio of *Q*_2_/*Q* keeps constant when the pump flow increases from the critical flow rate. In addition, whatever the flow rate of droplet generation is, the droplet is uniform, indicating strict requirements of pressure-controlled equipment will be released. In this method, the shear flow generation guarantees the uniformity of droplet size. Symmetric tubes change the way of producing droplets of dual pumps. The pump pressure affects the initial flow of fluid, and it determines whether the droplets are produced or not.

In this article, we presented and evaluated the potential of a capable droplet generator designed for practice in portable experimental studies. Droplet formation is usually governed by distinct “visco-capillary” regimes, rather than by a precise ratio between the flow rates of the two immiscible liquids in symmetric tubes. The droplet size is roughly determined by tube diameter, and is insensitive to the flow rate offered by the mono-pump. The flow rate determines the situation of droplets produced, including non-producing and defective droplets. A wide range of pump flow rates can be used in the droplet preparation process, which benefits from the shared pressure of the symmetric tube configuration. So, if we unbolt the first T-junction to the ambient atmosphere, they also share the same pressure. Opened symmetric tubes are constructed in the following experiment. As shown in Figure 5, the chip of the droplet generator from Bio-Rad Company which is used for QX200 Droplet Digital PCR was adopted for opened symmetric tubes to expand its applications, respectively. The two reservoirs of the chip store mineral oil and water phase, respectively. The droplet generator was assembled as shown in Figure 5a, and the pump was used to offer a flow rate. We observed that the microdroplets whose volume was at about 1nL were reliably produced with a relatively extreme difference in pump flow rate over 57.79% (see Appendix A). It is worth noting that only one flow pump was active during the whole process of droplet generation. This method ignores the pressure of the pump, so we will use a dropper instead of a mono-pump as the pressure source in the future. In Figure 5b, tens of thousands of uniform droplets were obtained in the reservoir of commercial components within a few seconds of releasing the hand from the dropper, with the RSD of the volume of droplets being about 9.80%. The rapidity and robustness of this principle of generating droplets are expected for a wide range of applications, especially for droplet-based bioanalysis, such as digital PCR, digital ELISA, etc. Such biological applications generally require rapidity and simultaneous droplet uniformity to increase the number of droplets and improve statistical analysis precision. Moreover, the droplet generator is less expensive, smaller, and lighter, which is more conducive to the portability of bioanalytical equipment, such as point-of-care tests.

## 5. Conclusions

Here, we describe a droplet generator design that aims to surpass existing systems in terms of portability, simplicity, and precision. In the experiment, once the pump flow is higher than the threshold, the droplets can be produced stably even when the pump flow rate is greater than 1.2 mL/min. Additionally, the volume of the micro-droplet is stable with a maximum error of 5%. If the rate is less than the threshold, the time interval of droplet generation will be too long, thus leading to a decrease in efficiency. The volume of the microdroplets can be regulated from 5 nL to 400 nL by changing the size of the structure in the junction. Such a design does not require high pump accuracy, but provides stable droplet generation with solid controllability. We replaced the pump with a dropper as a pressure source and obtained tens of millions of uniform droplets in the generator within a few seconds of releasing the hand from the dropper. We believe this capable droplet generator is suitable for portable bio-applications.

## Figures and Tables

**Figure 1 micromachines-14-00466-f001:**
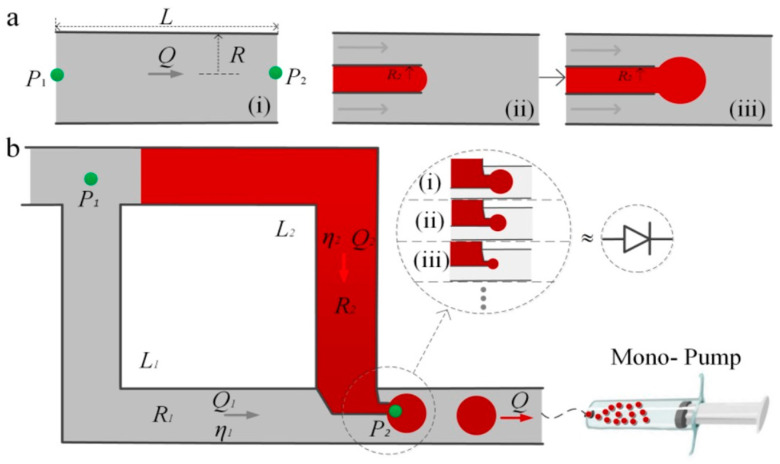
Schematic of the droplet generator. (**a**) Dynamic process of the droplet generator, (**i**) dimensional relationship in principles of Poiseuille law, (**ii**) water phase reached fluid tangent position, (**iii**) forming a droplet under the action of fluid shear force. (**b**) Schematic of droplet generator.

**Figure 2 micromachines-14-00466-f002:**
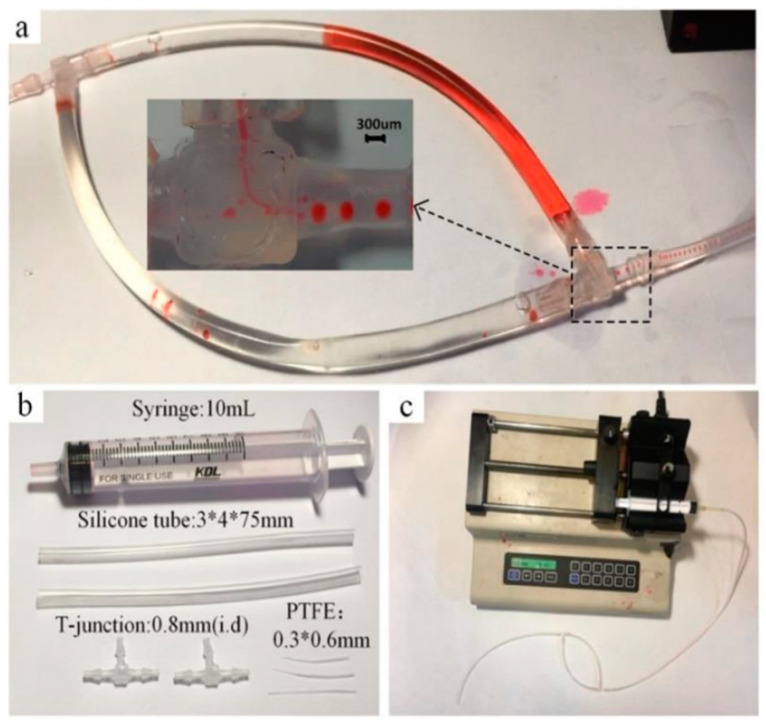
Component and assembly diagram of the droplet generator. (**a**) Shows the connection state of the symmetrical tubes during generation of droplets, (**b**) shows the core part of the material, (**c**) indicates the beginning state after connecting the syringe pump.

**Figure 3 micromachines-14-00466-f003:**
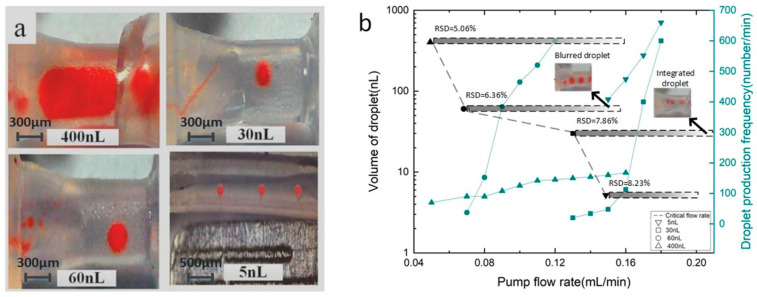
Study on droplet controllability. (**a**) Shows the droplet volumes were about 400 nL, 60 nL, 30 nL, and 5 nL, respectively, and (**b**) summarizes the critical flow rate of a droplet generator for four kinds of droplets.

**Figure 4 micromachines-14-00466-f004:**
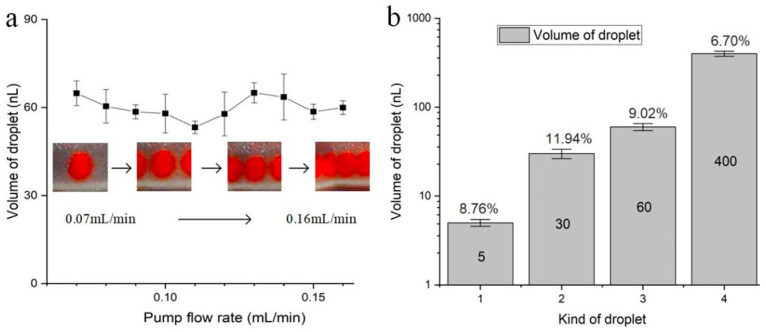
Stability and robustness of droplet size with ranging pump flow rate. (**a**) Shows the droplet volumes were about 60 nL with an increasing flow rate of a mono-pump from 0 mL/min, and (**b**) summarizes the RSD of the volume of four kinds of droplets.

**Figure 5 micromachines-14-00466-f005:**
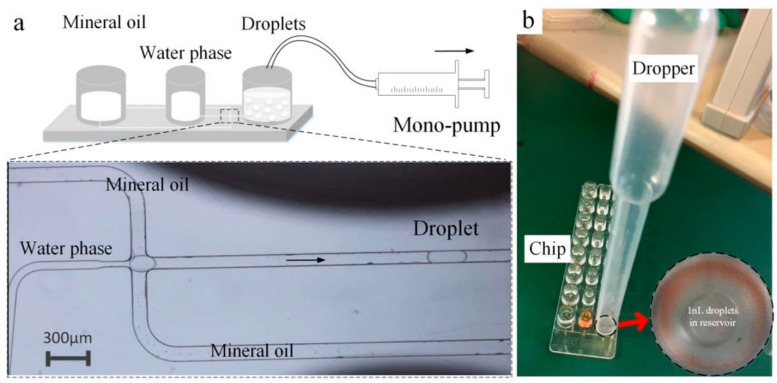
Stability and robustness of droplet size with ranging pump flow rate. (**a**) Shows the assembled droplet generator by the chip of droplet generator from Bio-rad Company, and (**b**) indicates obtained droplets in the reservoir of commercial components within a few seconds of releasing the hand from the dropper.

**Table 1 micromachines-14-00466-t001:** Table with the characteristics of the components.

Name	Number	Diameters
Silicone tube	4	id. 3 mm, od. 4 mm
T-junction	2	id. 0.8 mm
Teflon capillary tube	1	id. 0.3 mm, od. 0.6 mm

## Data Availability

The data supporting these study findings are available from the corresponding authors on reasonable requests.

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
