# Peer review of "Cost-Effective Droplet Generator for Portable Bio-Applications"

_micromachines, 2023, doi:10.3390/mi14020466_

Round 1

Reviewer 1 Report

1. Provide a clear description of your system

2. Eliminate jargon

3. Highlight novelty if any

4. Improve figures and photos

1. What is the main question addressed by the research? How to duplicate the function of a commercial instrument at a much reduced cost.

2. Do you consider the topic original or relevant in the field? Does it
address a specific gap in the field? The topic is not original. If there is a need, they could publish their work in their website or student conferences.

3. What does it add to the subject area compared with other published
material? Not much

4. What specific improvements should the authors consider regarding the
methodology? What further controls should be considered? The authors need to produce a complete system, show it works by conducting a PCR assay and provide a cost of the components and of the whole system.

5. Are the conclusions consistent with the evidence and arguments
presented and do they address the main question posed? They must build a complete system and show it works.

6. Are the references appropriate? Yes

7. Please include any additional comments on the tables and figures. The current figures are poor and the tables unreadable.

Author Response

Thanks a lot for the comments of the referees. The reply section is marked “A” in purple and the revised manuscript used review mode is uploaded.

Reviewer 2 Report

Du et al. present an experimental work on development of a droplet generator. the topic is highly investigated in literature, however there a quite new point of view is provided.

The manuscript needs an Extensive editing of English language and style . More sentences are inappropiate or/and unclear. (see, ad example lines: 31-32, 79, 80, 144-145 just to cite a few).

Lines 31  34,  99. The sentences "other scientists", "varius applications" are too generic and need to be better discussed.

The squeezing model theory should be better discussed. What is its  operation regime? Are you working in this regime for your esperiments? These points should be better clarified.

What do you mean for "capillary number"?

It is not clear the difference between section 2 and 3, in both of them you describe the generator. I suggest to not replicate the content of section 3 in section 2. Here, you should just describe the physics of the generator. in thissense, I halso suggest to change the title of section 2 in "Droplet generator principles". 

Line 140, it shoul be usefull add a table with all the characteristics of the tubes (e.g., the diameters)

Line 144. Add information about the pump. 

Line 146-147. Why these two liquids can simulate oil and water? please provide their viscosity and density .

Fig 2. The 3 figures need to be better described in the caption.

Fig 3b. I can not see the error bars. The quality of the figure is too low. Please, provide a description of Figs 3a and 3b. and Figs 4a and 4c in the captions

The conclusions are not really conclusions, but looks like  further discussions. This section need to be completely rewritten.

Minor issue: 

Add always a space between number and unit of measure.

Author Response

(The authors gave the same response as above.)

Round 2

Reviewer 1 Report

It can be accepted

Reviewer 2 Report

the authors have improved the paper and, in my opinion, it can be accepted for publication.